# The Study of Buckling and Post-Buckling of a Step-Variable FGM Box

**DOI:** 10.3390/ma12060918

**Published:** 2019-03-19

**Authors:** Leszek Czechowski, Zbigniew Kołakowski

**Affiliations:** Lodz University of Technology, 90-924 Łódź, Poland; zbigniew.kolakowski@p.lodz.pl

**Keywords:** finite element method, step-variable functionally graded materials, post-buckling state, thin-walled structures

## Abstract

This work concerns the analysis of a thin-walled box made of ceramic and step-variable functionally graded material (FGM) subjected to compression. The components of the box taken into account were pure alumina and aluminium-alumina graded material. The problem was solved on the basis of a finite element method and Koiter’s asymptotic theory using a semi-analytical method (SAM). It analysed both the buckling state and the post-buckling state of the box. In addition, three conditions were considered: The presence of alumina outside or inside of the box and a mixed case. The obtained results were presented and discussed.

## 1. Introduction

Since the concept of functionally graded material (FGM) was first presented in 1984 by Japanese researcher Niino, he and others dealt with the investigation of FGMs in the following years [1,2,3]. Nowadays, these types of materials are still treated as modern materials that, through varying different properties throughout their thickness, can carry loads in hard conditions, especially in high-temperature environment. The gradual changes in volume fraction of the components and non-homogenous structure allow continuous graded macroscopic properties to be obtained. At present, there are different techniques of producing FGMs—a gas-based method, liquid phase processes or solid phase processes [4]—but none of them provides perfect material distribution as it was given theoretically. Referring to the literature, one can find many papers devoted to the analysis of structures built of FGMs. Many of them concern the mainly theoretical study of structure behaviour especially under thermal, mechanical or mechanical-thermal loads. In a few works, analysis of FGM strength has been considered for thick elements (e.g., discs). Relating to the subject of the present paper, works based essentially on the study of stability are mentioned below. Bui et al. [5] investigated the plate subjected to a thermal environment using a novel third-order shear deformation plate theory. The authors of [6,7] studied the response of an alumina-steel plate under static or dynamic load on the basis of finite element method [6] and classic theory for a thin plate applying the Bubnov-Galerkin method and a four-order Runge-Kutta algorithm [7]. Trabelsi et al. [8] investigated the response of FGM shell structures (plates and cylindrical shells separately) due to a thermal load by the use of a first-order shear deformation theory (FOSDT). Reference [9] shows the study of a thick, simply supported FGM plate under bending on the grounds of displacement potential function (DPF). The authors of [10] applied two new higher-order transverse shear deformation theories (NHSDTs) to analyse the buckling and post-buckling states of an FGM plate. Xu et al. [11] examined the buckling behaviour of a rectangular plate on the basis of classic plate theory and the post-buckling state was verified by a means of ABAQUS software. The analysis of a plate under mechanical and thermal loads was included in Reference [12]. The authors solved the problem by using a nonlinear finite element method and a first-order shear plate theory. Liu, Ferreira, Xing and Neves [13] on the basis of a layer-wise shear deformation theory investigated the FGM sandwich shells and laminated composite shell. The layer-wise theory proposed by the authors was based on an expansion of Mindlin’s first-order shear deformation theory (FOSDT) in each layer. Mechab et al. [14] presented analytical solutions of FGM plates on the basis of a four-variable refined theory. The authors of [15] investigated the influence of the imperfection sign on the local and post-buckling equilibrium path of an FGM plate on the basis of Koiter’s theory. The results received from a semi-analytical method (SAM) were compared to those of a finite element method (FEM) score. Tung and Duc [16] studied the behaviour of an FGM plate under in-plane compressive, thermal and combined loads based on classic plate theory (CPT). Yang and Shen [17] also analysed an FGM plate under thermo-mechanical loads regarding various boundary conditions. They used Reddy’s higher-order shear deformation plate theory (HOSDPT). The researchers of [18,19,20] studied the buckling and post-buckling states of FGM columns on the basis of classical laminate plate theory (CLPT). The analyses of stability of composite plate structures regarding experimental and numerical results were widely carried out in [21,22,23,24]. Pedersen [25] analysed on the basis of some structure the extension from the elasticity results to nonlinearity case. Lellap and Majak [26] focused on the problem of optimal material orientation using nonlinear elastic material. The same authors in [27] solved the problem of the minimization of elastic energy density for nonlinear elastic solids. In Reference [28], the elastic stability of solid structures was studied. This book is not limited to buckling or linear instability but also contains the theory of nonlinear post-buckling behaviour and imperfection sensitivity. Monograph [29] was concerned with the buckling and post-buckling behavior of thin plates and thin-walled structures with a flat wall subjected to both static and dynamic loads. The book of Hui-Shen [30] was devoted to a study of the geometrically nonlinear response of inhomogeneous isotropic or functionally graded plates and shells. Iha et al. [31] conducted a critical review of reported studies in recent years for a range of thermo-elastic and vibrations analyses of FGMs and structures. The authors of [32] focused on the review of a mesh-less method in the analysis of composite materials for plate and shell structures. Reddy [33] reported on the basis of the third-order shear deformation plate theory and examined the through-thickness of functionally graded plates. Various methods of study relating to the static, dynamic and stability behavior of FGM plates were collected in [34]. Liang et al. [35] used in their analysis the Koiter–Newton approach to model the buckling of geometrically nonlinear structures.

The authors of the present paper focused on the study of an alumina-FGM (alumina-alumina-aluminium) box with a finite number of layers. The box of 11 layers with step-variable gradation across the wall was assumed to relate to a real FGM structure manufactured by different techniques. In addition, an analysis was carried out to assess the behaviour of the structure when the layer of ceramic is inside or outside of all walls, or when ceramic layers exist on two opposite walls inside of the box and metal is on the two other walls. The initial imperfection in all cases was equal to 0.01 of wall thickness corresponding to first mode of buckling. Moreover, in this work the phenomenon of post-critical equilibrium paths for the considered cases was deeply explained. Owing to the fact that on the basis of literature, a similar analysis of step-variable gradation material in connection with ceramic on such a box has not been conducted before, the results of the present study seem to be very interesting especially as they separately consider two methods of calculation. 

## 2. Problem Description

The object of investigation was a box subjected to a compression load (Figure 1). The length and total thickness of the box amounted to a = b = c = 200 mm and $t_{t}$ = 2 mm, respectively. The thicknesses of pure ceramic $t_{c}$ and $t_{FGM}$ were equal to 0.2 mm and 1.8 mm, respectively, where $t_{t}=t_{FGM}+t_{c}$. The entire thickness was comprised of FGM (Al-Al_2_O_3_ with 11 layers) and pure ceramics Al_2_O _3_ (Figure 2). The material properties of the analysed box were based on two basic materials, Al and Al_2_O _3_, as given in Table 1. Material properties for the remaining layers in the FGM (as in Figure 2) were determined using the mixture law of two components. 

### 2.1. Finite Element Model (FEM)

Numerical simulation on the basis of the finite element method was performed in Ansys 18.2^®^ software (2018) [36]. To create an appropriate numerical model a 281-shell element was assumed. This finite element possessed eight nodes with six degrees of freedom in each of them and was suitable to analyse moderately thick shell structures. With the use of that element one can perform linear and nonlinear simulation of multilayered structures including a sandwich structure. The accuracy of modelling composite materials by means of a 281-shell element was based on a first-order deformation theory (FODT) (it usually refers to Mindlin-Reissner’s shell theory). The plate box was divided into 40,000 finite elements (100 elements along the edge—Figure 3a). 

The nonlinear analysis for large strain and deflections was conducted by using Green–Lagrangian equations. To ensure the appropriate equilibrium, simulations were executed with compliance to the Newton–Raphson algorithm. The initial imperfection in all cases was assumed to be $0.01t_{t}$ which referred to the first buckling mode. The exemplary setup of layers in the finite element (FE) model (corner of the box) is illustrated in Figure 3b (in this case the “mixed” variant is shown). Boundary conditions in the FE model reflected to articulated supports on all edges (Table 2, ●—yes, ○—no).

### 2.2. Koiter’s Asymptotic Approach

The nonlinear problem of stability has been solved with the semi-analytical method (SAM) based on Koiter’s nonlinear theory [28,29,37,38]. The full Green’s strain tensor, the second Piola-Kirchhoff’s stress tensor and the transition matrix using Godunov’s orthogonalisation have been used in the description of the problem [18,19,37,38]. In Reference [15], on the basis of Koiter’s theory, the FG plates have non-symmetric stable post-buckling equilibrium paths. This feature explains the differences in the plate response dependence on the imperfection sign (sense). An FGM plate has a non-trivial coupling matrix B and the coupling between extensional and bending deformations exists as in the case of non-symmetric laminated plates [15]. The equilibrium equations for FGM plate structures can be written as [15,18]:(1)$$\left(1-\frac{\sigma }{\sigma_{r}}\right)\zeta_{r}+a_{pqr}\zeta_{p}\zeta_{q}+b_{rrrr}\zeta_{r}^{3}-\frac{\sigma }{\sigma_{r}}\zeta_{r}^{*}+...=0\;\mathrm{for}\;r=1,.....J.$$

In References [15,28,29], on the basis of Koiter’s theory, the FGM plates have non-symmetric stable post-buckling equilibrium paths. This feature explains the differences in the plate response dependence on the imperfection sign (direction of deflection). 

An FGM plate has a non-trivial coupling matrix B and the coupling between extensional and bending deformations exists as it in the case of non-symmetric laminated plates [15], where $\sigma_{r}$—critical stress instead of the *r**-*th buckling mode, $\zeta_{r}$—dimensionless amplitude of the *r*-th buckling mode, $\zeta_{r}^{*}$—dimensionless amplitude of the initial imperfections corresponding to the *r-*th buckling mode, σ—compressive stress, $a_{apq}$ and $b_{rrrr}$—coefficients. The range of indices (p, q, r) is from 1 to J, where J is the number of interacting modes. The summation is supposed on the repeated indices. For the case of the uncoupled buckling mode (i.e., for one mode approach) J=1 must be used. The first-order coefficients (i.e., $a_{apq}$) are found with the analytical-numerical method (ANM) based on Koiter’s theory [15,28,38]. The second-order coefficients (i.e., $b_{rrrr}$) are calculated with the semi-analytical method (SAM) [37]. In this method, one postulates to determine approximated values of $b_{rrrr}$ on the basis of the linear buckling problem. It is worth mentioning that the semi-analytical method (SAM) allows phenomena to be analysed and interpreted in a considerably simpler way as compared to the results obtained by FEM.

## 3. Results and Discussion

Before starting with the analysis of the results, one should return to Reference [15], where compliance with Koiter’s theory enabled the phenomenon of the influence of the imperfection sign on local post-critical equilibrium paths of plates made of functionally graded material to be explained. For the plate structures built of FGM, non-symmetric stable post-critical equilibrium paths were obtained. In the study, two methods were employed: SAM and FEM. Results received from FEM are for geometrically complete nonlinear analysis. However, in SAM nonlinear expressions linked with imperfection were omitted but values of post-critical coefficients of the second order were approximately determined. This causes essential differences in the results obtained by two methods, SAM and FEM, for the same value of imperfection. It was revealed that an initial deflection in ceramic direction gives higher values of total deflections for a given value of structure load as compared to an initial deflection in metal direction for an inflicted absolute value of imperfection. For FEM, differences in deflections in metal and ceramic direction are more visible than in case of SAM. The deflection in ceramic direction relates to greater stability, whereas a deflection in metal direction corresponds to weaker stability. By retaining greater stability we obtained a lower level of total system energy as compared to that achieved with lower stability. Real systems always trend to the lowest energy level. In the present work, results were shown for a compressed column with the shape of a hexahedron in which either outer layer of the detailed plates is made of ceramic or made of aluminium. For both cases, small differences in critical load and post-critical equilibrium paths for loads close to critical loading were obtained.

### 3.1. Buckling Forces

On the basis of SAM and FEM according to boundary conditions (Table 2 and Figure 3a), critical forces for three first modes were obtained (Table 3). Differences in buckling forces for the first mode between FEM and SAM amounted to 1.67%, 2.51% and 0.55% for variant_1, variant_2 and variant_3, respectively. SAM gave lower forces in the first two cases. Furthermore, as it can be easily noticed, for a mixed arrangement of material the critical force appeared to be the lowest one for FEM and for SAM between for variations 1 and 2. For higher modes of buckling forces, differences in both methods did not exceed 1.2%, at most. The first modes attained by applying FEM turned out to be the same regardless of ceramic position (Figure 4). Regarding linear buckling, the maximum deflection was always noticed on the wall in the y-direction.

### 3.2. Post-Buckling State

The phenomenon can be clarified in the following way. In the case of an isotropic cubic hexahedron, the cross-section area possesses two symmetric axes and two anti-symmetric axes (this means those axes are diagonals of the quadratic cross-section). In the case of an FGM box with a cube shape relating to conclusions from [15], we have only two symmetric axes but the diagonals of the cross-section are not anti-symmetric axes. Let us consider an FGM cube whose detailed plates in the corner are connected by an articulated joint. In this case the corners of the box do not carry bending moments and right angle is not retained. As far as such a box whose outer layer is ceramic can be considered, then according to conclusions of [15], the post-critical deflection of all plates will be outside the box. This is called blowing from the inside or the inflation of the box. If the inner layer of the box is ceramic then post-critical deflections of all plates are directed toward the inside. In the present work it also assumed for the cube with different arrangements in neighbouring plates (a so-called “mixed” cube, which denotes a structure of one plate with an outer ceramic layer and a second neighbouring plate with an aluminium layer—Figure 1c) that the arrangement of the walls responds to the buckling of all the plates in the ceramic direction. This causes the post-critical equilibrium paths for all plates to be the lowest in comparison to the two remaining cases (ceramic inside or outside). This is due to the greater stability of the equilibrium path or the deepening influence of the initial deflection. Figure 5, Figure 6, Figure 7 and Figure 8 show the deflections in the centre of each plate of the box vs. static load ($F_{st}$). For wall 1 (Figure 5), curves obtained both by FEM and by SAM were posted. For the remaining walls, only results from FEM were displayed because the deflections received by SAM for four walls differed slightly from each other (approximately 0.1%). The characteristic for variant_2 (FEM) is the closest to the curves of SAM but deflections (for SAM) seem to be greater (10%–15%, at most) especially over the critical load. As seen in Figure 5, regardless of the considered variant, trends of SAM coincide with each other. Owing to this fact, the main analysis was focused only on FEM results.

In the cases of ceramic outside and ceramic inside, the post-critical paths are very close to each other for walls 1, 2 and 4 (see Figure 5, Figure 6 and Figure 8). The exception is the behaviour of wall 3 (Figure 7). This is caused by much smaller differences in amplitudes of mode deflection for detailed walls (amounting to 0.001). Attention should be paid to the course of the curves for the ceramic inside in Figure 5 and Figure 6 and for the ceramic outside in Figure 7 and Figure 8. In practice, until reaching a critical value, curves are the straight lines at minimal deviation into reverse direction with regard to post-critical paths for load over the critical load. Those distances correspond to deflections in the direction at a lower stability of equilibrium paths, following which small deflections pass through towards the greater stability of paths. For higher values of initial deflections (imperfections), the phenomenon may occur for bigger overloads above the critical load or be completely absent.

## 4. Conclusions

In this paper, the results of the semi-analytical method (SAM) and the finite element method (FEM) employed to analyse the structure of an alumina-FGM plate subjected to static load were presented. The behaviour of a box built of ceramic and step-variable graded material with a finite number of layers was investigated. Three models were assumed: ceramic inside, ceramic outside and a mixed presence of ceramic. Based on the obtained results, the following can be concluded:(1)For variant_3 (with a mixed arrangement of material) the first critical force was the lowest. For higher critical forces, differences were not noticeable and for FEM they did not exceed 1%. Furthermore, in this case deflections on all walls appeared the fastest in comparison to other considered options, but finally at greater overload curves were grew closer to each other.(2)The difference in the deflection of walls achieved by SAM amounted to 0.1% in the whole scope and this does not give substantial contrast in comparison to analysed variants as it was observed in FEM.(3)The SAM allows results of equilibrium paths to be achieved significantly faster than FEM. SAM is very useful for interpreting the phenomena accompanying the interaction of different modes of buckling in the full scope of the load. This method furthers our understanding of phenomena that occur during coupled buckling.

## Figures and Tables

**Figure 1 materials-12-00918-f001:**
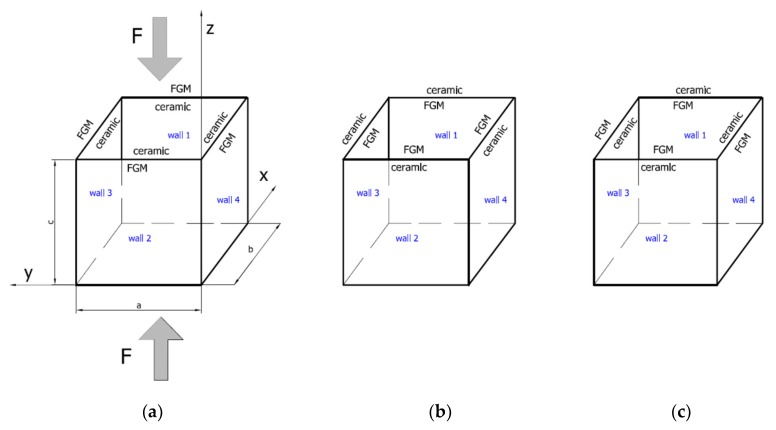
The functionally graded material (FGM) box with its dimensions and coordinate system: Ceramic inside (**a**: variant_1), ceramic outside (**b**: variant_2) and “mixed” case (**c**: variant_3).

**Figure 2 materials-12-00918-f002:**
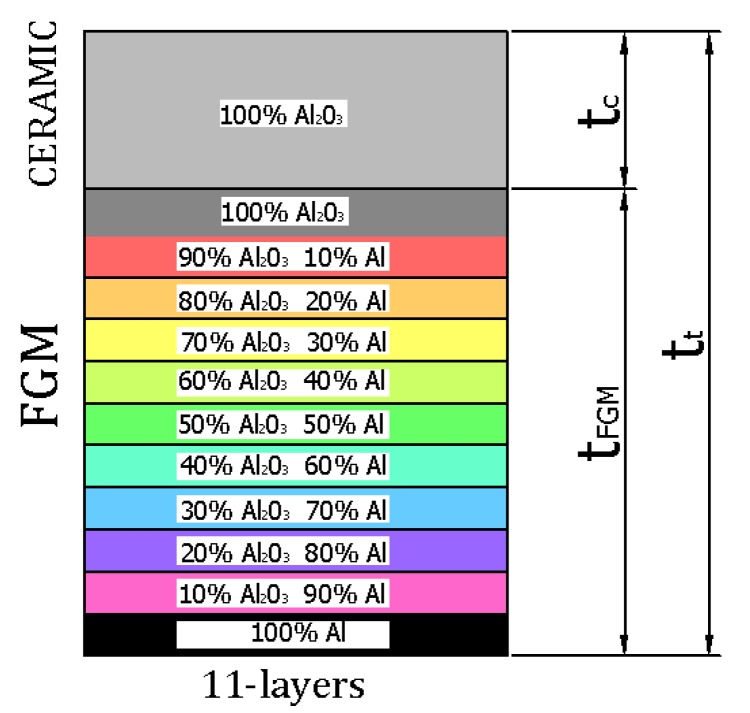
Material distribution across the box thickness.

**Figure 3 materials-12-00918-f003:**
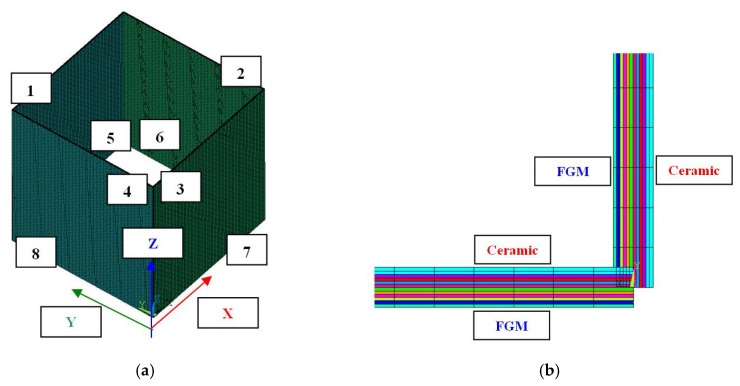
Discrete model (**a**) and distribution of material for variant_3 (**b**).

**Figure 4 materials-12-00918-f004:**
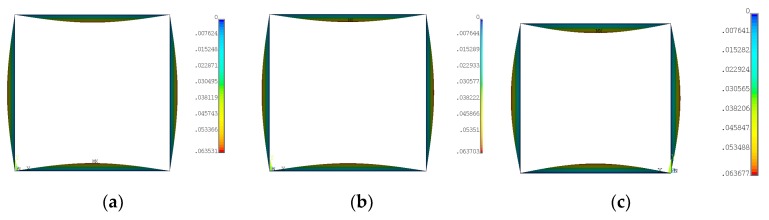
First buckling mode for variant_1 (**a**), for variant_2 (**b**) and for variant_3 (**c**).

**Figure 5 materials-12-00918-f005:**
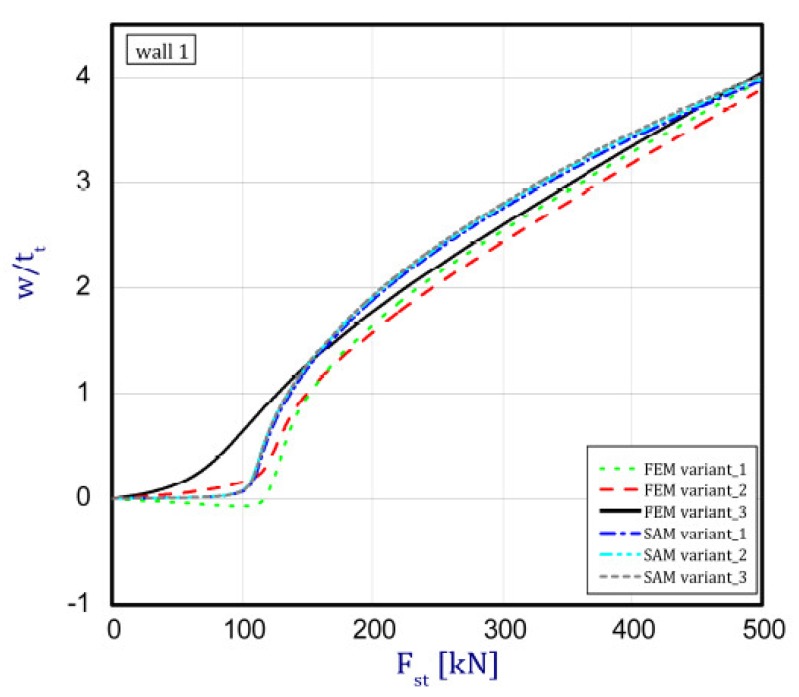
Deflection of wall 1 vs. load.

**Figure 6 materials-12-00918-f006:**
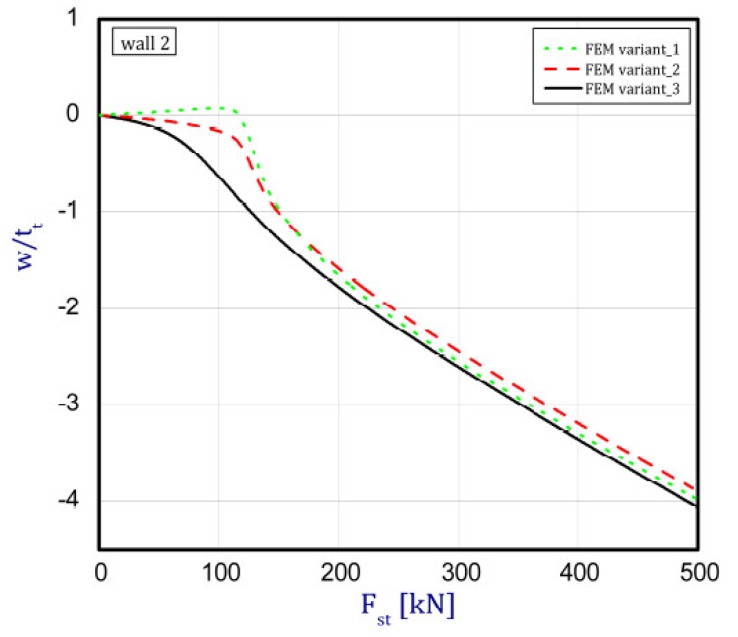
Deflection of wall 2 vs. load.

**Figure 7 materials-12-00918-f007:**
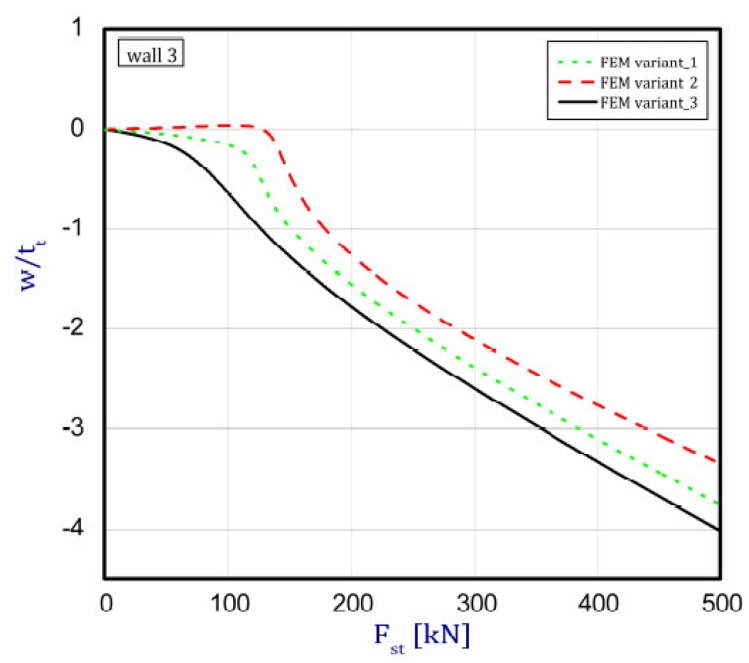
Deflection of wall 3 vs. load.

**Figure 8 materials-12-00918-f008:**
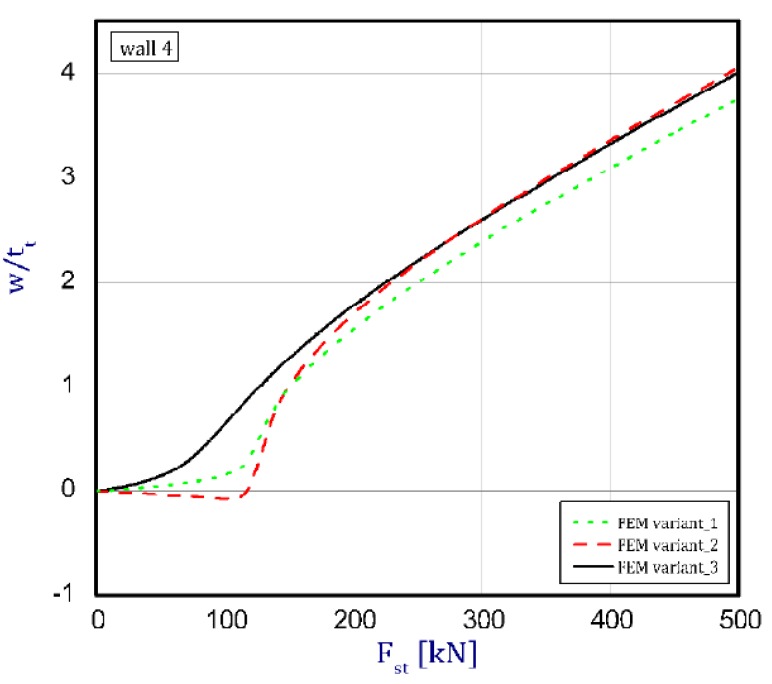
Deflection of wall 4 vs. load.

**Table 1 materials-12-00918-t001:** Material properties of basic constituents.

Components	Young Modulus (GPa)	Poisson’s Ratio (-)
Al	70	0.33
Al_2_O_3_	393	0.25

**Table 2 materials-12-00918-t002:** Assumed boundary conditions in finite element (FE) model according to Figure 3.

Number of Edges	u_x_	u_y_	u_z_	rot_x_	rot_y_	rot_z_	Couple Degree of Freedom in Nodes in the z-Direction	Load in the z-Direction
1	○	●	const	○	○	○	●	●
2	●	○	const	○	○	○	●	●
3	○	●	const	○	○	○	●	●
4	●	○	const	○	○	○	●	●
5	○	●	●	○	○	○	○	○
6	●	○	●	○	○	○	○	○
7	○	●	●	○	○	○	○	○
8	●	○	●	○	○	○	○	○

**Table 3 materials-12-00918-t003:** Critical forces in kN.

Method of Solution	Buckling Mode	Variant_1	Variant_2	Variant_3
Finite element method (FEM)	First	115.490	114.961	112.176
	Second	162.128	163.014	162.669
	Third	177.761	177.344	177.056
Semi-analytical method (SAM)	First	113.564	112.076	112.787
	Second	162.331	161.046	162.452
	Third	176.572	175.507	176.014

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
