# Peer review of "The Study of Buckling and Post-Buckling of a Step-Variable FGM Box"

_materials, 2019, doi:10.3390/ma12060918_

Round 1
Reviewer 1 Report
The nonlinear analysis for large strain and deflections was conducted by using 92 Green-Lagrangian equations. Small differences in critical load and post-critical equilibrium paths for loads close to critical loading were observed.
Authors contribution exists. Suggestion , accept with minor revision.
Remarks:
1. Page 5: „Differences in buckling forces for first mode between FEM and SAM amounted to 1.67%, 2.51%, 0.55% for variant_1, variant_2 and variant_3, respectively.“
Very good agreement achieved. What about forces for second (or higher) modes? Here are often/typically accuracy problems, how in your case.
2. The problem considered is complex due to geometrical non-linearity considered. However, in range of large deflections, strains, the linear elastic constitutive model appear not good enough. What about application of the proposed approach to various nonlinear elastic constitutive models like:
Pedersen P., Some general optimal design results using anisotropic, power law nonlinear elasticity, Structural Optimization 15, 73-80, 1998.
Lellep, J. Majak, J. Nonlinear constitutive behavior of orthotropic materials. Mechanics of Composite Materials,2000, 36 (4), 261−266.
Lellep, J. Majak, J. On optimal orientation of nonlinear elastic orthotropic materials. Structural Optimization, 1997, 14, 116−120.
Is proposed approach directly applicable, need some modification or significant adaption. This can be discussed in introduction.
3. The literature review is generally adequate. However, currently there is 1/3 of authors papers (these papers are related to topic and showing authors experience in field, can be kept). The literature review should be completed by adding 10-20 papers of other authors (nonlinear material models mentioned, etc.) in order to keep normal proportion of authors work.
Author Response
Dear Reviewer,
thank you for review of our manuscript.
The responses to your questions/remarks were included in attached file.
The text of manuscript was corrected according to your suggestions.
With best Regards,
L. Czechowski
Z. Kołakowski

Reviewer 2 Report
The paper aims to investigate the buckling and post-buckling behaviors of thin-walled boxes made of ceramic and step-variable functionally graded material (FGM). A semi-analytical approach is used to this aim. A major revision is suggested due to the following considerations:
1. The literature review is not complete. The topic of FGM is extremely wide (several applications concerning the buckling analysis of FGM structures have been already published), therefore both the introduction and the list of references should be improved accordingly.
2. In general, an FGM is characterized by a continuous gradual variation of the volume fraction of both the constituents. According to figure 2, it seems that the structure is characterized by 11 layers with defined mechanical properties (gradual, but not continuous). Please, clarify this aspect.
3. Further detail concerning the SAM could be added in the manuscript.
4. Some typos should be corrected throughout the whole manuscript.
5. The conclusions could be improved by highlighting more the novelties and the main results of the paper. In addition, the sentence “This section may be divided by subheadings” in the same paragraph has no sense (probably it is a typo).
6. In general, the novelty of the paper could be more emphasized throughout the whole manuscript.
Author Response

(The authors gave the same response as above.)

Round 2
Reviewer 2 Report
The paper has noticeably improved by the authors following the suggestions provided by the reviewers. The following final observation could be taken into account. In the reviewer's opinion, the fact that the structures under consideration are made of step-variable FGM could be emphasized also in the title for the sake of clarity, since this aspect is quite important. Therefore, the title of the research could be modified consequently. For instance, the following one could be used: "The study of buckling and post-buckling of step-variable FGM box".
Author Response
Dear Reviewer,
once again thank you for review and valuable suggestions/remarks.
With best Regards,
L. Czechowski
Z. Kołakowski
